# Stable, Fast and Accurate: Kernelized Attention with Relative Positional Encoding

**Shengjie Luo**[1*], **Shanda Li**[2*], **Tianle Cai**[4], **Di He**[6†],
**Dinglan Peng**[5], **Shuxin Zheng**[6†], **Guolin Ke**[5], **Liwei Wang**[1,2,3], **Tie-Yan Liu**[6]
[1]Center for Data Science, Peking University
[2]Key Laboratory of Machine Perception, MOE, School of EECS, Peking University
[3]Institute for Artificial Intelligence, Peking University    [4]Princeton University
[5]University of Science and Technology of China    [6]Microsoft Research
luosj@stu.pku.edu.cn, lishanda@pku.edu.cn,
tianle.cai@princeton.edu, pengdinglan@mail.ustc.edu.cn,
{dihe, shuz, guoke, tyliu}@microsoft.com, wanglw@pku.edu.cn

## Abstract

The attention module, which is a crucial component in Transformer, cannot scale efficiently to long sequences due to its quadratic complexity. Many works focus on approximating the dot-then-exponentiate softmax function in the original attention, leading to sub-quadratic or even linear-complexity Transformer architectures. However, we show that these methods cannot be applied to more powerful attention modules that go beyond the dot-then-exponentiate style, e.g., Transformers with relative positional encoding (RPE). Since in many state-of-the-art models, relative positional encoding is used as default, designing efficient Transformers that can incorporate RPE is appealing. In this paper, we propose a novel way to accelerate attention calculation for Transformers with RPE on top of the kernelized attention. Based upon the observation that relative positional encoding forms a Toeplitz matrix, we mathematically show that kernelized attention with RPE can be calculated efficiently using Fast Fourier Transform (FFT). With FFT, our method achieves $\mathcal{O}(n \log n)$ time complexity. Interestingly, we further demonstrate that properly using relative positional encoding can mitigate the training instability problem of vanilla kernelized attention. On a wide range of tasks, we empirically show that our models can be trained from scratch without any optimization issues. The learned model performs better than many efficient Transformer variants and is faster than standard Transformer in the long-sequence regime.

## 1 Introduction

Transformer [45] has achieved tremendous success on a variety of tasks in natural language processing [34, 9, 50], computer vision [2, 10, 42] and speech [13]. Despite the great success, Transformers are usually inefficient in modeling long-sequence input since the key component—the attention module—needs to calculate pairwise correlations between all the positions, which comes with quadratic time and memory cost with respect to the sequence length $n$.

To reduce the cost, many works focus on approximating the dot-then-exponentiate function in the attention module of the original Transformer [41]. Most works can be roughly categorized into three classes: sparse attention [3, 33, 30, 15, 21, 46, 7], low-rank approximation-based attention [49, 48] and kernelized attention [17, 32, 4]. Sparse attention variants design either pre-defined

---

[*]Equal contribution.
[†]Correspondence to: Di He <dihe@microsoft.com> and Shuxin Zheng <shuz@microsoft.com>.

35th Conference on Neural Information Processing Systems (NeurIPS 2021).

[3, 33, 30, 15] or learnable patterns [21, 46, 7] to reduce the amount of the key-value pairs that each query needs to attend to. Low-rank approximation-based attention variants project the key matrix into a length-agnostic low-dimensional space [48], or use Nyström approximation [49] to reduce the quadratic dependency on sequence length. The kernelized attention is first developed in [44, 17], which introduces a kernel view on the dot-then-exponentiate softmax function and comes up with a linear attention mechanism. Followed-up works [32, 4] theoretically investigate the choices of different kernel feature maps and propose several random feature maps that achieve much better performance on various applications.

However, in recently developed Transformers, the attention mechanism is designed to be more complicated than dot-then-exponentiation. For example, the relative positional encoding (RPE) [38, 35] has become standard in state-of-the-art models [35, 18, 14]: Besides using the dot-products between queries and keys, RPE incorporates a position-correlation matrix in the softmax exponentiation to encode the distance between any two positions. With RPE, Transformers can effectively capture the relative word orders and achieve superior performance on various tasks. As Transformer with RPE is more powerful, one may wonder whether the model can be still approximated by directly applying previous acceleration approaches.

Unfortunately, we mathematically show that such a term expands the expressiveness of dot-then-exponentiation, i.e., for some attention with RPE, there exists no dot-then-exponentiate function that can represent it. Therefore, previous approaches cannot be directly applied to approximate Transformers with RPE and obtain non-trivial speed-up. In this paper, we develop a novel attention computation that achieves an almost-linear speed-up rate for Transformer with RPE. We build our method on top of the kernelized attention in [4, 32]. The kernelized attention method replaces the dot-then-exponentiate attention by a simple matrix multiplication between the kernelized features of queries and key-value pairs. When taking RPE into the kernelized attention, one can naively multiply the $n \times n$ RPE position-correlation matrix with kernelized feature matrices, which leads to a trivial quadratic time cost. While interestingly, we find that the elements in the RPE position-correlation matrix are determined by the relative distance between positions, and this matrix is precisely a Toeplitz matrix. In numerical linear algebra, it is well-known that multiplication between a Toeplitz matrix and row vectors can be accelerated using Fast Fourier Transform (FFT). And by using FFT, we can obtain a fast attention computation with $\mathcal{O}(n \log n)$ time cost.

Beyond the efficiency, we also find that our method that incorporates RPE into kernelized attention can also stabilize the training of such models. As demonstrated by several works [16, 4], although these kernelized attention methods have theoretical guarantees to approximate the softmax attention, they face optimization issues and will diverge when training from scratch for large-scale tasks. This significantly limits the practical impact of the model as we have to train a standard Transformer first and convert it to kernelized attention with finetuning. We first shed light upon the reason for this phenomenon by proving that these kernelized attentions have an unacceptable variance when the $\ell_2$ norms of queries and keys are large. However, queries and keys with large norms usually appear during training, leading to optimization instability or sub-optimal solutions. When incorporating RPE, we can properly normalize queries and keys to obtain a "low-variance" attention, making the training stable. Since the value range of the relative positional encodings is not constrained, the whole model can still have enough capacity to achieve good performance.

We conduct experiments on a wide range of tasks spanning language pre-training, machine translation and image classification. On all the tasks, we show that our model can be trained from scratch stably, outperforms other competitive efficient models, and is faster than the standard Transformer in the long-sequence regime.

## 2 Preliminary

### 2.1 Attention Module and its Kernel View

The attention module is one of the key components in the Transformer [45, 9]. It is usually formulated as querying a dictionary with key-value pairs, e.g., Attention$(Q, K, V) =$ softmax $\left( \frac{QW^Q(KW^K)^\top}{\sqrt{d}} \right) VW^V$, where $d$ is the dimensionality of the hidden representations. $Q$ (Query), $K$ (Key), $V$ (Value) are specified as the hidden representations of the previous layer and $W^Q, W^K$ and $W^V$ are projection matrices for any specific head.

To be more concrete, we denote $x = (x_1, x_2 \cdots, x_n)$ as the input to the attention module, where $n$ is the length of the sequence and $x_i \in \mathbb{R}^d$ is a row vector denoting the contextual representation of the token at position $i$. Denote $z = (z_1, z_2 \cdots, z_n)$ as the output. Then the attention module can be written as

$$z_i = \sum_{j=1}^{n} \frac{\exp(\alpha_{ij})}{\sum_{j'=1}^{n} \exp(\alpha_{ij'})}(x_j W^V), \text{ where } \alpha_{ij} = \frac{1}{\sqrt{d}}(x_i W^Q)(x_j W^K)^\top. \tag{1}$$

As we can see, the calculation of attention for each position requires $\mathcal{O}(n)$ time and memory cost. Thus, the time and memory cost for the whole sequence is $\mathcal{O}(n^2)$, which impedes the attention mechanism scaling to long sequences. Recently, [44, 17] introduce a kernel view on the attention mechanism and derive a general kernelized attention framework as follows:

$$z_i = \sum_{j=1}^{n} \frac{\kappa(x_i W^Q, x_j W^K)}{\sum_{j'=1}^{n} \kappa(x_i W^Q, x_{j'} W^K)}(x_j W^V), \tag{2}$$

where $\kappa(\cdot, \cdot) : \mathbb{R}^d \times \mathbb{R}^d \to \mathbb{R}$ is any positive-definite kernel measuring the pairwise similarity. Based on this kernel view, the attention mechanism can be alternatively modeled as a linear dot-product of kernelized features:

$$z_i = \sum_{j=1}^{n} \frac{\phi(x_i W^Q)\phi(x_j W^K)^\top}{\sum_{j'=1}^{n} \phi(x_i W^Q)\phi(x_{j'} W^K)^\top}(x_j W^V) = \frac{\phi(x_i W^Q) \sum_{j=1}^{n} \phi(x_j W^K)^\top (x_j W^V)}{\phi(x_i W^Q) \sum_{j'=1}^{n} \phi(x_{j'} W^K)^\top}, \tag{3}$$

where $\phi(\cdot) : \mathbb{R}^d \to \mathbb{R}^m$ is the feature map and $m$ is the dimension of the feature space. Since both $\sum_{j=1}^{n} \phi(x_j W^K)(x_j W^V)$ and $\sum_{j'=1}^{n} \phi(x'_j W^K)$ can be reused for each position, the time and memory complexity of the kernelized attention is reduced to $\mathcal{O}(n)$. [17] simply uses $\text{elu}(\cdot) + 1$ as $\phi(\cdot)$. Followed-up works [32, 4] further propose two (randomized) feature maps with which the kernels are unbiased estimators of the dot-then-exponentiate function, therefore achieve similar expressiveness as the standard Transformer. For simplicity, we call the feature maps used in [32] and [4] as Trigonometric Random Feature (TRF) and Positive Random Feature (PRF) respectively:

$$\phi_{TRF}(x) = \frac{\exp(\frac{\|x\|^2}{2})}{\sqrt{m}} \left[ \sin(w_1 x^\top), ..., \sin(w_m x^\top), \cos(w_1 x^\top), ..., \cos(w_m x^\top) \right], \tag{4}$$

$$\phi_{PRF}(x) = \frac{\exp(-\frac{\|x\|^2}{2})}{\sqrt{m}} \left[ \exp(w_1 x^\top), ..., \exp(w_m x^\top) \right], \tag{5}$$

where each vector $w_i$ is independently sampled from $N(0, I_d)$. Our work aims to design an efficient Transformer with relative positional encoding on top of the kernelized attention. Without loss of generality, we use Positive Random Features as the feature map to demonstrate the effectiveness of our approach. Our approach can also be applied to other feature maps such as the Trigonometric Random Features. See Section 4.5 for some ablation studies.

## 2.2 Relative Positional Encoding

Relative Positional Encoding (RPE) is recognized as one of the most successful modifications to the original Transformer model [27]. RPE is first introduced in [38], which suggests that using absolute positional encoding may be ineffective in capturing relative word orders. They propose an embedding matrix in which the values of the elements are determined by the distance between the row index and column index. [35] further proposes a simple yet more effective relative positional encoding, which is popularly used in many recent state-of-the-art Transformers, including [1, 18, 24]. In this model [35], the relative positional encoding is served as a bias term added to the dot-product between queries and keys:

$$\alpha_{ij}^{rel} = \frac{1}{\sqrt{d}}(x_i W^Q)(x_j W^K)^\top + b_{j-i}. \tag{6}$$

For each $j - i$, $b_{j-i}$ is a learnable scalar and shared across all layers.

## 3 Incorporating RPE into Kernelized Attention

We are curious about whether previous acceleration methods can be directly applied to Transformers with RPE for achieving both speed-up and better accuracy. In this section, we first provide a negative result: We prove that RPE expands the expressiveness of the original attention module. So those efficient approaches that can only approximate dot-then-exponentiate functions *cannot* approximate Transformers with RPE at least in the sense of representation power. To address the problem, we develop new techniques for computing attention with RPE on top of the kernelized attention, and show that we can still obtain non-trivial speed-up using Fast Fourier Transform. Lastly, we study how to stabilize the training of the PRF attention and achieve good performance with the help of RPE. Due to space limitations, we put all the proofs in Appendix B.

### 3.1 Attention with RPE Goes Beyond the Dot-then-exponentiate Function Class

For simplicity, we study the single-head attention as a demonstration where $W_{rel}^Q, W_{rel}^K \in \mathbb{R}^{d \times d}$. All the conclusions remain the same for the multi-head attention setting. The following proposition shows that Transformers with RPE are strictly more expressive than those without RPE.

**Proposition 1.** *If $n > d+1$, there exist weight matrices $W_{rel}^Q, W_{rel}^K \in \mathbb{R}^{d \times d}$ and RPE weights $b_{j-i}$ as in Equation 6, such that there does **not** exist weight matrices $W_{vanilla}^Q, W_{vanilla}^K$ and input vectors $x_1, \cdots, x_n \in \mathbb{R}^d$ that satisfy*

$$\frac{\exp(\alpha_{ij}^{rel})}{\sum_{k=1}^n \exp(\alpha_{ik}^{rel})} = \frac{\exp(\alpha_{ij}^{vanilla})}{\sum_{k=1}^n \exp(\alpha_{ik}^{vanilla})}, \ \forall i, j. \tag{7}$$

*where*

$$\alpha_{ij}^{rel} = \frac{1}{\sqrt{d}}(x_i W_{rel}^Q)(x_j W_{rel}^K)^\top + b_{j-i}; \tag{8}$$

$$\alpha_{ij}^{vanilla} = \frac{1}{\sqrt{d}}(x_i W_{vanilla}^Q)(x_j W_{vanilla}^K)^\top. \tag{9}$$

This theoretical finding is consistent with all empirical observations that Transformers equipped with RPE are more powerful and usually perform better than the original Transformer. At the same time, such a result suggests that Transformers with RPE cannot be well-approximated by techniques developed in the previous works as those methods are under the simple dot-then-exponentiate attention setting. To accelerate Transformers with RPE, new techniques need to be explored.

### 3.2 Fast Attention Calculation using Fast Fourier Transform

We propose an efficient method that accelerates the computation of attention modules with RPE. The approach is built on top of the kernelized attentions, i.e., Equation 3. When relative positional encoding is further implemented, the kernelized attention will be formulated as:

$$z_i = \frac{\phi(x_i W^Q) \sum_{j=1}^n e^{b_{j-i}} \phi(x_j W^K)^\top (x_j W^V)}{\phi(x_i W^Q) \sum_{j=1}^n e^{b_{j-i}} \phi(x_j W^K)^\top}. \tag{10}$$

From Equation 10, we can see that to calculate the output $(z_i)_{i=1}^n$, we need to compute $D_1 = \left(\sum_{j=1}^n e^{b_{j-i}} \phi(x_j W^K)^\top (x_j W^V)\right)_{i=1}^n$ and $D_2 = \left(\sum_{j=1}^n e^{b_{j-i}} \phi(x_j W^K)\right)_{i=1}^n$. To better illustrate the calculation step, we reshape $D_1, D_2$ into the form where elements are vectorized as row vectors and stacked into matrices:

$$\tilde{D}_1 = \begin{pmatrix} \text{vec}\left(\sum_{j=1}^n e^{b_{j-1}} \phi(x_j W^K)^\top (x_j W^V)\right) \\ \vdots \\ \text{vec}\left(\sum_{j=1}^n e^{b_{j-n}} \phi(x_j W^K)^\top (x_j W^V)\right) \end{pmatrix}, \tilde{D}_2 = \begin{pmatrix} \text{vec}\left(\sum_{j=1}^n e^{b_{j-1}} \phi(x_j W^K)^\top\right) \\ \vdots \\ \text{vec}\left(\sum_{j=1}^n e^{b_{j-n}} \phi(x_j W^K)^\top\right) \end{pmatrix},$$

$$\tag{11}$$

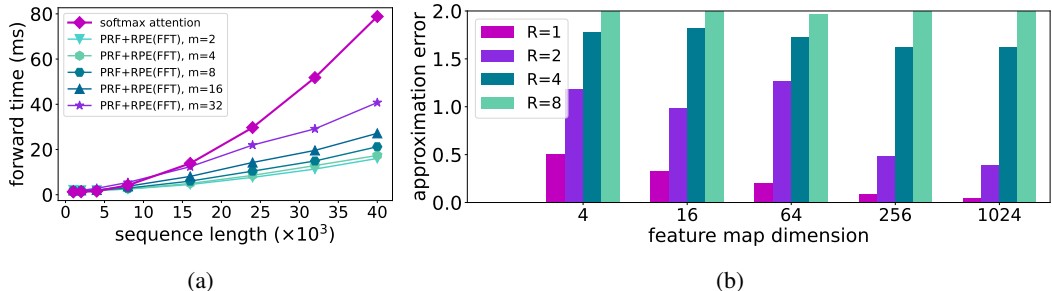

Figure 1: (a). Model forward speed of vanilla Transformer and our model with different feature map dimensions ($m$). (b). Approximation error of PRF with different query/key norms ($R$).

where $\mathrm{vec}\,(\cdot)$ denotes vectorization operation. Let $c_i = \mathrm{e}^{b_i}$. $\tilde{D}_1, \tilde{D}_2$ can be factorized as the multiplication of two known matrices:

$$\tilde{D}_1 = \begin{pmatrix} c_0 & c_1 & c_2 & \cdots & c_{n-1} \\ c_{-1} & c_0 & c_1 & \cdots & c_{n-2} \\ \vdots & \vdots & \vdots & \ddots & \vdots \\ c_{-(n-1)} & c_{-(n-2)} & c_{-(n-3)} & \cdots & c_0 \end{pmatrix} \begin{pmatrix} \mathrm{vec}\left(\phi(x_1 W^K)^\top (x_1 W^V)\right) \\ \mathrm{vec}\left(\phi(x_2 W^K)^\top (x_2 W^V)\right) \\ \vdots \\ \mathrm{vec}\left(\phi(x_n W^K)^\top (x_n W^V)\right) \end{pmatrix}, \quad (12)$$

$$\tilde{D}_2 = \begin{pmatrix} c_0 & c_1 & c_2 & \cdots & c_{n-1} \\ c_{-1} & c_0 & c_1 & \cdots & c_{n-2} \\ \vdots & \vdots & \vdots & \ddots & \vdots \\ c_{-(n-1)} & c_{-(n-2)} & c_{-(n-3)} & \cdots & c_0 \end{pmatrix} \begin{pmatrix} \mathrm{vec}\left(\phi(x_1 W^K)^\top\right) \\ \mathrm{vec}\left(\phi(x_2 W^K)^\top\right) \\ \vdots \\ \mathrm{vec}\left(\phi(x_n W^K)^\top\right) \end{pmatrix}. \quad (13)$$

Denote $C = \{c_{j-i}\}_{i,j=1}^{n}$ as the positional-correlation matrix. It can be easily seen that one can use naive matrix multiplication to obtain $\tilde{D}_1, \tilde{D}_2$. But it leads to quadratic time cost with respect to the sequence length $n$ and makes the advantage of using kernelized attention disappear. Interestingly, we notice that $C$ is a diagonal-constant matrix in which each descending diagonal from left to right is constant, i.e., a Toeplitz matrix. In numerical linear algebra [11], left multiplying a Toeplitz matrix can be done in $\mathcal{O}(n \log n)$ time using Fast Fourier Transform (FFT). This fact leads to an efficient algorithm to compute kernelized attention with RPE in $\mathcal{O}(n \log n)$ time. [3]

We empirically evaluate the forward cost of the kernelized Transformer with RPE using FFT and the standard Transformer. For simplicity, we set the number of heads to 1 and the size of embedding to 64. We use synthetic input sequences with lengths ranging from 1k to 40k. For our model, we experiment with different feature map dimensions. All the experiments are run on a Tesla V100 GPU with 16GB memory for ten trials. The averaged result is shown in Figure 1a. It can be seen that while the forward cost of standard attention becomes much significant on longer sequences, our model still remains efficient (almost linear) in the long-sequence regime.

### 3.3 RPE Enables Stable Training of Kernelized Attention

Recent works [4, 32] show that PRF and TRF attention can approximate the dot-then-exponentiate attention with theoretical guarantees and achieves much better performance than other methods. However, the training of these models is challenging. As demonstrated by several works [16, 4], kernelized attention can hardly be optimized from scratch for large-scale tasks and may even diverge during training. This significantly limits the practical impact of the model as one has to train a standard Transformer first and convert it to the kernelized version. [16] further observed that the model conversion usually leads to a considerable performance drop which suggests that the

---

[3]One limitation of the proposed method is that it cannot accelerate sequence generation models *during the inference step*. In our approach, $D_1$ and $D_2$ need to be pre-computed using FFT for all the queries as a whole. However, in the inference step of the sequence generation, we cannot pre-compute $D_1$ and $D_2$ as the tokens are generated on the fly and the FFT operation needs to be used in each time step. But our method can still be used to accelerate the training of the generation model which uses teacher forcing by setting $b_{i-j} = -\infty$ (i.e., $c_{i-j} = 0$) when $i > j$. We leave how to accelerate the inference step using our method as future work.

approximation seems to be inaccurate. To understand the problem, we start with Lemma 2 below from [4], which analyzes the variance of the softmax-kernel estimation with the PRF feature map.

**Lemma 2** (Variance of the PRF (Lemma 2 in [4])). *For $\phi_{PRF}$ defined in Equation 5, we have*

$$\mathrm{Var}(\phi(x)\phi(y)^{\top}) = \frac{1}{m}\left(\exp(\|x+y\|^2) - 1\right)\exp(xy^{\top})^2. \tag{14}$$

It can be seen from the above lemma that the variance of the estimation is related to the *scale of the queries and keys*. If the scale of the queries and keys is large, the variance will be significant. Our following theorem precisely characterizes that one may need an exponential number of feature map dimensions to obtain a well-approximated PRF kernelized attention. Similar results also hold for the TRF kernel [4].

**Theorem 3** (Sample Complexity of Approximation). *For a query $q$ and $n$ keys $k_1, \cdots, k_n$, let $A$, $\hat{A}$ be the attention score obtained by the standard softmax and the PRF approximation respectively. Assume that $\ell_2$-norms of the query and keys are upper-bounded by $R$. Then for any $\varepsilon > 0$, if feature map dimension $m = \Theta(\frac{n\exp(4R^2)}{\varepsilon^2\delta})$, we have $\Pr\left(\|A - \hat{A}\|_1 \geq \varepsilon\right) \leq \delta$.*

We also perform numerical simulations to verify the theoretical findings. We set $d = 64$ as the query/key dimension and independently sample a query vector and 1024 key vectors from the uniform distribution over a $d$ dimensional unit hypersphere. Therefore, the $\ell_2$-norm of the query and the keys we sampled equals one. Denote $R \geq 1$ as a scaling factor. For each value of $R$, we rescale the query and the keys with $R$ and calculate the softmax score $A$ and the approximated score $\hat{A}$ using PRF with different feature map dimensions. We use $\|A - \hat{A}\|_1$ to quantify the approximation error. The results are shown in Figure 1b. It can be seen that when $R$ is large, the approximation error is huge and can only be slightly improved as the feature map dimension grows from 4 to 1024. But when $R = 1$, the approximation error is small and drops significantly when we increase the feature map dimension. This numerical result is consistent with our theory, which implies that the scale of the queries and keys significantly matters the approximation quality.

Based on the results, we can understand why the model conversion (well-trained standard attention $\rightarrow$ kernelized attention) usually leads to performance drop as observed in [16]. For a well-trained Transformer, some attention heads generate sharp attention distributions (e.g., attending to the next token [6]). To generate a sharp attention distribution, queries and keys with a large norm are essential. If we convert those attention heads to their kernelized version, the output can be significantly different due to the large variance, and the performance will be poor. Furthermore, queries and keys with a large norm may also be the root cause of the training instability [4]. As discussed in [45], at initialization, the parameters are sampled from Gaussian distribution and unbounded. If the norms of the queries or keys happen to be larger than one, the kernelized attention (forward pass) will have a large variance, leading to a large-variance gradient (backward pass) and making the training unstable.

**Normalized Kernelized Attention with RPE.** A straightforward solution to fix the problem is normalizing queries and keys by their $\ell_2$ norms[5], so the variance of the kernelized attention will be well controlled. However, this would bound the dot-products between queries and keys into range $[-1, 1]$. Then the attention mechanism can no longer represent sharp distributions, which will hurt the model performance. Fortunately, when equipped with RPE, the model is endowed with the ability to model sharp attention distributions even if the queries and keys are normalized, since relative positional encoding $b_{j-i}$ remains unbounded. For ease of reference, we call our model Normalized PRF Transformer with RPE and use NPRF-Transformer with RPE for short. A pseudocode implementation is provided in Algorithm 1. In the next section, we will show that

---

[4]In [4], the authors provide a sample complexity bound of the approximation error between the TRF kernel and the dot-then-exponentiate function without considering the denominator term in the softmax. In this paper, we directly provide the sample complexity bound on the approximation error between the PRF-induced attention distribution and the softmax attention distribution.

[5]The TRF paper [32] also proposes to normalize queries and keys. In [32], the normalized queries and keys are multiplied with a Gaussian vector $w$ which is *rescaled* using a learnable vector $\sigma \in \mathbb{R}^d$. This is equivalent to rescaling the normalized $q$ and $k$ using $\sigma$ and then feeding them into $\phi(\cdot)$. This doesn't control the norms of queries and keys since the scale of $\sigma$ can be large, which doesn't address the optimization issue.

**Algorithm 1** Efficient Normalized Kernelized Attention with RPE using FFT

**function** RPE_NKA( $Q$, $K$, $V$, $W^Q$, $W^K$, $W^V$, $b = (b_{-(n-1)}, \cdots, b_{n-1})$, $\phi$):

    \# $X_i$ denotes the $i$-th element of $X$

    $Q \leftarrow \left( \frac{Q_i W^Q}{\|Q_i W^Q\|_2} \right)_{i=1}^n, K \leftarrow \left( \frac{K_i W^K}{\|K_i W^K\|_2} \right)_{i=1}^n, V \leftarrow \left( V_i W^V \right)_{i=1}^n$

    $c \leftarrow \exp(b), A_1 \leftarrow \left( \phi(K_i)^\top V_i \right)_{i=1}^n, A_2 \leftarrow \left( \phi(K_i)^\top \right)_{i=1}^n$

    \# FFTMatrixMul() is an FFT-based algorithm for multiplying a Toeplitz
    matrix as in Eq. (12), (13). We omit the reshaping steps and assume the
    second input variable and the output have the same shape.

    $D_1 \leftarrow$ FFTMatrixMul$(c, A_1), D_2 \leftarrow$ FFTMatrixMul$(c, A_2)$

    $X \leftarrow \left( \frac{\phi(Q_i) D_{1,i}}{\phi(Q_i) D_{2,i}} \right)_{i=1}^n$

    **return** $X$.

**end**

Table 1: GLUE scores on dev set. "*" indicates the best performance.

| Method | Complexity | CoLA | RTE | MRPC | STS-B | MNLI | QNLI | QQP | SST-2 | AVG |
|---|---|---|---|---|---|---|---|---|---|---|
| *Finetune from RoBERTa-base* | | | | | | | | | | |
| SMYRF [7] | $\mathcal{O}(n \log n)$ | 58.8 | 68.6 | 87.7 | 89.7 | 85.0 | 91.1 | 89.7 | 93.2 | 83.0 |
| FCA [46] | $\mathcal{O}(n)$ | 59.8 | 49.8 | 43.6 | 78.9 | 79.4 | 74.6 | 89.4 | 94.4 | 71.2 |
| Improved FCA [46] | $\mathcal{O}(n)$ | 60.1 | 70.4 | 87.3 | 90.0* | 88.0* | 93.0* | 91.5 | 94.7 | 84.4 |
| *Pre-train from scratch* | | | | | | | | | | |
| Linformer [48] | $\mathcal{O}(n)$ | / | / | / | / | / | 91.2 | 90.8 | 93.1 | / |
| Nyströmformer [49] | $\mathcal{O}(n)$ | / | / | 88.1 | / | 80.9 | 88.7 | 86.3 | 91.4 | / |
| Ours | $\mathcal{O}(n \log n)$ | 64.7* | 75.1* | 88.5* | 89.4 | 86.4 | 91.3 | 91.7* | 94.8* | 85.2* |

NPRF-Transformer with RPE can achieve comparable performances with vanilla Transformers and can be trained from scratch for all tasks due to using the low-variance kernels. [6]

## 4 Experiments

In this section, we conduct experiments to verify the effectiveness of our approach on several benchmark datasets covering language pre-training, language modeling, machine translation, and image classification. Then we provide the ablation study and discussions on the design choices. Due to space limitations, detailed description of the experiment settings are presented in Appendix A.

### 4.1 Language Pre-training

We use the BERT-base architecture [9] in our experiments, which consists of 12 Transformer layers. For each layer, the hidden size is set to 768, and the number of attention heads is set to 12. We follow [23] to construct the data corpus and train the model using masked language modeling task. We use the GLUE (General Language Understanding Evaluation) dataset [47] as the downstream tasks to evaluate the performance of the pre-trained models. All codes are implemented based on *fairseq* [28] in *PyTorch* [31]. All models are run on 64 NVIDIA Tesla V100 GPUs with mixed-precision [26].

To show the effectiveness of our proposed NPRF-Transformer with RPE, we choose several competitive baselines in literature, including : Linformer [48], Nyströmformer [49], SMYRF [7], and Fast Clustered Attention (FCA) [46]. All the baselines use the same architecture as ours. The details

---

[6]The expressive power of NPRF-Transformer with RPE can be weaker than vanilla Transformer as it can only produce *context-agnostic* sharp distribution (from RPE) but may not produce *context-aware* sharp patterns. However, this does not conflict with the observation that NPRF-Transformer with RPE outperforms PRF-Transformer in practice. Although PRF Transformer seems to have more flexibility, its training instability issue hinders the model from learning well.

Table 2: Language model perplexity scores on WikiText-103 validation set. We use $^*$ to indicate the best performance. All the results of the baseline methods are reported in [32].

| Model | Perplexity |
|---|---|
| Vanilla Transformer | 33.0 |
| Linear Transformer | 38.4 |
| TRF-Transformer | 33.6 |
| TRF-Transformer-GATE | 31.3 |
| NPRF-Transformer w/ RPE (Ours) | 30.6$^*$ |

about these baselines can be found in Appendix A.1. For all baselines, we report the number in their original papers. We also tried to train the PRF Transformer model from scratch or continue to train a PRF Transformer from the released RoBERTa checkpoint [23]. But all the efforts failed due to the optimization instability issue. For our model, we directly modify the self-attention layer using normalized kernelized attention with RPE and train it from scratch.

The overall comparison results are presented in Table 1 ("/" indicates the GLUE score of the corresponding downstream task is not reported). It can be easily seen that our NPRF-Transformer with RPE outperforms the previous works in terms of the average GLUE score, with a light computational overhead. Beyond the superior performance, it is also worth noting that our model can be stably pre-trained from scratch, which is particularly important in practice.

## 4.2 Language Modeling

We conduct experiments on WikiText-103 language modeling task to demonstrate the effectiveness of our proposed method. We compare our model with the following baselines: 1. Transformer with softmax attention. 2. *Linear Transformer* proposed in [17], which uses kernelized attention with $\mathrm{elu}(\cdot) + 1$ as its feature map. 3. TRF-Transformer using TRF kernel (called *RFA* in [32]). 4. *TRF-Transformer-GATE* [32], which uses a gating mechanism to further boost performance.

Following [32], the sequence length is set to 512 during both training and evaluation. All models are trained without access to the context from previous mini-batches for a fair comparison. The model architecture consists of 6 decoder layers. The number of attention head is set to 8. The hidden dimension is set to 512. The dimension of feed-forward layer is set to 2048. The dropout ratio and the weight decay are set to 0.1 and 0.01, respectively. The batch size is set to 64. The feature map dimension is set to 64. We use Adam [19] as the optimizer, and set its hyperparameter $\epsilon$ to $1\mathrm{e}-6$ and $(\beta_1, \beta_2)$ to $(0.9, 0.98)$. The peak learning rate is set to $2\mathrm{e}-3$. The model is trained for 150k steps with a 6k-step warm-up stage followed by an inverse square-root learning rate scheduler. All models are trained on 8 NVIDIA Tesla V100 GPUs.

The results are shown in Table 2. Our NPRF-Transformer with RPE outperforms all the baselines by a large margin, e.g., it achieves a 30.6 perplexity score which is 3.0 lower than the baseline TRF-Transformer and 0.7 lower than TRF-Transformer-GATE. It is worth noting that we mainly make architectural changes and should compare our model with TRF-Transformer. In [32], a set of training tricks (such as gating) are introduced for improving the performance. Our method can be combined with those tricks for further improvements.

## 4.3 Machine Translation

In machine translation, we evaluate our method on the widely used public dataset IWSLT14 German↔English and French↔English. For each language pair, we use a vocabulary of 10K tokens based on a joint source and target byte pair encoding (BPE) [36]. All the experiments use a Transformer encoder-decoder architecture, with a 6-layer encoder and 6-layer decoder. We set the embedding dimension to 512 and the number of heads to 4. For a fair comparison, we set the feature map dimension to 16 in the encoder and 24 in the decoder whenever kernelized attention is used. BLEU [29] is used as the evaluation measure of the model performance. In this small-scale task, we find the optimization for all the model variants is stable, and all models are trained from scratch.

Table 3: Test BLEU scores on machine translation tasks. We use */** to indicate the best performance obtained by the standard model / efficient model. "Enc/dec" denotes "encoder/decoder" for short.

| Transformer model type | Complextity encoder / decoder | Encoder softmax PRF | | Decoder softmax PRF | | BLEU de-en | en-de | fr-en | en-fr | avg |
|---|---|---|---|---|---|---|---|---|---|---|
| Standard enc-dec | $\mathcal{O}(n^2)/\mathcal{O}(n^2)$ | ✓ | | ✓ | | 34.7* | 29.2 | 40.0 | 40.2 | 36.0 |
| Standard enc + PRF dec | $\mathcal{O}(n^2)/\mathcal{O}(n)$ | ✓ | | | ✓ | 34.6 | 29.3* | 40.3* | 40.6* | 36.2* |
| PRF enc-dec | $\mathcal{O}(n)/\mathcal{O}(n)$ | | ✓ | | ✓ | 32.1 | 27.6 | 38.0 | 38.2 | 34.0 |
| NPRF enc-dec w/ RPE (Ours) | $\mathcal{O}(n \log n)/\mathcal{O}(n \log n)$ | | ✓ | | ✓ | 34.3** | 29.4** | 39.9** | 40.4** | 36.0** |

Table 4: Test accuracy on ImageNet. We use */** to indicate the best performance obtained by the standard model / efficient model.

| Model | Complexity | Query/key standard | normalized | Attention softmax | PRF | ImageNet Top-1 acc | Top-5 acc |
|---|---|---|---|---|---|---|---|
| ViT-base[10] | $\mathcal{O}(n^2)$ | ✓ | | ✓ | | 77.9 | / |
| ViT-large[10] | $\mathcal{O}(n^2)$ | ✓ | | ✓ | | 76.5 | / |
| DeiT-base | $\mathcal{O}(n^2)$ | ✓ | | ✓ | | 81.2* | 95.0* |
| PRF DeiT-base (fine-tuned) | $\mathcal{O}(n)$ | ✓ | | | ✓ | 79.5 | 94.1 |
| NPRF DeiT-base w/o RPE | $\mathcal{O}(n)$ | | ✓ | | ✓ | 77.7 | 93.3 |
| NPRF DeiT-base w/ RPE (Ours) | $\mathcal{O}(n \log n)$ | | ✓ | | ✓ | 80.9** | 94.8** |

As the translation task uses an encoder-decoder framework, it is natural to check whether the attention modification (i.e., kernelized attention) can be applied to both encoder and decoder. In [32], the authors only show that using TRF attention in the decoder will not hurt the model performance. But we argue that a long target sequence usually corresponds to a long source sequence in translation, and modifying the attentions in both encoder and decoder should be equally important.

We give a systematic study on the model performance of using PRF in the encoder/decoder and summarize the results shown in Table 3. We first observe that models using PRF kernel in the decoder are comparable with or even slightly better than standard ones (the 2nd line v.s. the 1st line), which is consistent with [32]. The performance improvement may be attributed to regularization effects brought by the randomness in PRF. However, the overall time cost should still be $\mathcal{O}(n^2)$ as the standard attention is used in the encoder. When applying the kernelized attention to both encoder and decoder, we observe a significant performance drop (the 3rd line v.s. the 1st line). This may indicate that when using more layers of kernelized attention, the model becomes harder to train due to accumulated approximation error through layers. On the contrary, our method can incorporate kernelized attention in both encoder and decoder, achieving the same average BLEU score as the standard Transformer.

## 4.4 Image Classification on ImageNet

Beyond the natural language tasks, we also extend our study to image classification. We use the DeiT-base [42] as our backbone architecture, which consists of 12 Transformer layers. For each layer, the hidden size is set to 768, and the number of attention heads is set to 12. We set the feature map dimension to 32 whenever kernelized attention is used. Following [10, 42], we benchmark our method on ImageNet-1K [8], which contains 1.28M training images and 50K validation images from 1,000 classes. For evaluation, the top-1/top-5 accuracy on a single crop is reported. For our proposed method, we use the 2-dimensional relative positional encoding developed in [24].

We compare our proposed model with several baseline models using standard attentions, including ViT-base/ViT-large proposed in [10] and the DeiT-base [42]. We also compare our model with a finetuned PRF attention model from DeiT, and an NPRF attention model without RPE. The result is shown in Table 4. First, we can see that our model achieves competitive performance compared to standard attention models (81.2 v.s. 80.9 in terms of top-1 accuracy, 95.0 v.s. 94.8 in terms of top-5 accuracy). Second, we can see that compared with other efficient variants, our model achieves the best performance, and the use of normalized queries/keys and RPE are both helpful.

## 4.5 More Analyses

We conducted more studies the IWSLT14 German-to-English (De-En) task to ablate our designs.

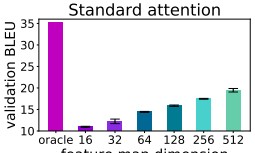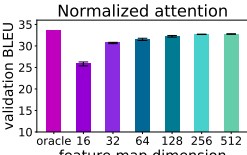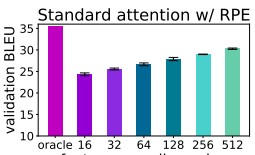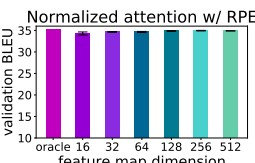

Figure 2: Performance of kernelized attention models converted from well-trained Transformers. "Oracle" represents the BLEU score of the well-trained model for reference.

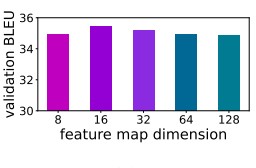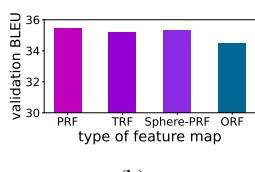

(a)                                    (b)

Figure 3: (a). Performance of the NPRF-Transformer with RPE using different feature map dimensions. (b). Performance of the models using different feature map functions.

**Normalized attention is easy to be converted to its kernelized version**    We conduct experiments to study whether using normalized attention can practically mitigate the performance drop issue observed in the previous works. We train four model variants: Transformer with standard/normalized attention and with/without RPE. After the training finishes, we replace the attention softmax functions by Equation 5 *without* further finetuning. For each setting, we run with 5 random seeds. Figure 2 summarizes the experiment results in terms of the validation BLEU scores with confidence intervals.

We can see that Transformer with standard attention suffers a considerable performance degradation after the conversion. In contrast, Transformer with normalized attention only suffers a small performance drop. This clearly shows that normalization reduces the approximate error significantly. Interestingly, we observe that RPE *universally* improves the performance of the converted model for both standard and normalized attention. For our NPRF-Transformer with RPE, the feature map dimension is not sensitive any longer, e.g., the converted model only has a 0.5 performance drop when the feature map dimension is only 32.

**Affects of feature map dimensions and other choices of feature map**    The dimension of feature map $\phi$ is a hyper-parameter that balances the accuracy and efficiency. For our NPRF-Transformer with RPE, we train models from scratch with different feature map dimensions and check the validation BLEU. We can see from Figure 3a that the feature map dimension is not sensitive regarding the final performance. Setting the dimensionality to 16 is even a slightly better choice.

Our approach is general which can be applied to any feature map function. The main body of the paper showcases models using the PRF feature map, and here we provide some empirical comparisons with other feature maps. Specifically, we experiment with 1) the TRF feature map defined by Equation 4; 2) the Sphere-PRF feature map, which is designed in [4] as an alternative choice of PRF, where all $w_i$ in Equation 5 are sampled independently from $\mathrm{Unif}(\sqrt{d}\mathbb{S}^{d-1})$; 3) the ORF feature map, which is also designed in [4] where $w_1, \cdots, w_D$ in Equation 5 are forced to be orthogonal to each other. For all the choices, we train models with normalized kernelized attention and RPE under the same setting as in Section 4.3 and report the best validation BLEU. The results are shown in Figure 3b. We can see that when we normalize the attention and use RPE, all models obtain similar performance.

## 5    Conclusion

In this paper, we propose a novel way to accelerate attention calculation for Transformers with relative positional encoding and achieve $\mathcal{O}(n \log n)$ time complexity. Our method is built on top of the kernelized attention. Using the fact that relative positional encoding forms a Toeplitz matrix, we mathematically show that kernelized attention with RPE can be calculated efficiently using Fast Fourier Transform (FFT). We further demonstrate the additional benefit of using relative positional encoding from the optimization perspective. As mentioned in the main paper, we will extend the current method to the decoder model in generation tasks and explore more strategies to accelerate attention in the long-sequence regime.

## Acknowledgements

We thank all the anonymous reviewers for their constructive comments. This work was supported by National Key R&D Program of China (2018YFB1402600), Key-Area Research and Development Program of Guangdong Province (No.2019B121204008), BJNSF (L172037) and Beijing Academy of Artificial Intelligence. Project 2020BD006 supported by PKU-Baidu Fund.

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
