covering Low-Rank Approximation based Attention, Sparse Attention and Kernelized Attention, including : 1. *Linformer* proposed in [48], which projects the length dimension of keys and values to lower-dimensional representations ($N \rightarrow K$) and induces $\mathcal{O}(NK)$ time complexity; 2. *Nyströmformer* proposed in [49], which uses Nyström approximation and reduces the time complexity to $\mathcal{O}(NM)$, where $M(M \ll N)$ is the number of landmarks of Nyström method; 3. *SMYRF* proposed in [7], which uses Locality Sensitive Hashing (LSH) and newly defined Asymmetric transformation to produce balanced query-key clusters, introducing $\mathcal{O}(N \log N)$ time complexity; 4. *Fast Clustered Attention (FCA)* proposed in [46], which first groups queries into clusters and computes attention for the centroids, inducing $\mathcal{O}(NC)$ time complexity with $C$ denoting the number of clusters.

**Pre-training.** Following [23], we collect five English-language corpora of varying sizes and domains, totaling over 160GB of uncompressed text. The pre-training corpora consist of Bookcorpus[51] and English wikipedia, CC-News[7], OpenWebText[8] and Stories[43]. Detailed description of these corpora can be found in [23]. We follow [23] to conduct a couple of consecutive pre-processing steps: segmenting documents into sentences by Spacy[9], normalizing, lower-casing, and tokenizing the texts by using SentencePiece [22].

We use masked language modeling as the objective of pre-training. We train the models for 1000k steps where the batch size is 2048 and the maximum sequence length is 512. The masked probability is set to 0.15, and we replace 80% of the masked positions with `[MASK]`, 10% with randomly sampled words, and keep the remaining 10% unchanged. We use Adam [19] as the optimizer, and set its hyperparameter $\epsilon$ to $1e-6$ and $(\beta_1, \beta_2)$ to (0.9, 0.999). The peak learning rate is set to $2e-4$ with a 20k-step warm-up stage. After the warm-up stage, the learning rate decays linearly to zero. We set the dropout probability to 0.1, gradient clip norm to 1.0, and weight decay to 0.01. All models are trained on 64 NVIDIA Tesla V100 GPUs. The pre-training setting details are listed in Table 5.

**Fine-tuning.** We use the GLUE (General Language Understanding Evaluation) dataset [47] as the downstream tasks to evaluate the performance of the pre-trained models. Particularly, we use nine tasks in GLUE, including CoLA, RTE, MRPC, STS, SST, QNLI, QQP, and MNLI. For the evaluation metrics, we report Matthews correlation for CoLA, Pearson correlation for STS-B, and accuracy for other tasks. We use the same optimizer (Adam) as in pre-training. Following previous works, we search the learning rates during the fine-tuning for each downstream task. For a fair comparison, we do not apply any tricks for fine-tuning. Each configuration will be run five times with different random seeds, and the median of these five results on the development set will be used as the performance of one configuration. We report the best score over all configurations. The fine-tuning setting details are listed in Table 5.

## A.2 Language Modeling

**Baselines.** We compare our model with the following baselines: 1. *Transformer* with softmax attention. 2. *Linear Transformer* proposed in [17], which uses kernelized attention with $elu(\cdot) + 1$ as its feature map. 3. TRF-Transformer using TRF kernel (called *RFA* in [32]). 4. *TRF-Transformer-GATE* [32], which uses a gating mechanism to further boost performance.

**Settings.** Following [32], the sequence length is set to 512 during both training and evaluation. All models are trained without access to the context from previous mini-batches for a fair comparison. The model architecture consists of 6 decoder layers. The number of attention head is set to 8. The hidden dimension is set to 512. The dimension of feed-forward layer is set to 2048. The dropout ratio

---

[7] http://web.archive.org/save/http://commoncrawl.org/2016/10/newsdataset-available

[8] http://web.archive.org/save/http://Skylion007.github.io/OpenWebTextCorpus

[9] https://spacy.io

[10] We use five for the top four high-resource tasks, MNLI-m/-mm, QQP, and QNLI, to save the fine-tuning costs. Ten is used for other tasks.

Table 5: Hyperparameters used in the Language Pre-training task.

| | Pre-training | Fine-tuning |
|---|---|---|
| *Model Configuration* | | |
| **Layers** | | 12 |
| **Hidden Representation** | | 768 |
| **Feed-Forward Layer** | | 3072 |
| **Heads** | | 12 |
| *Hyperparameters* | | |
| **Max Steps** | 1000k | - |
| **Max Epochs** | - | 5 or 10 [10] |
| **Learning Rate** | 2e-4 | {5e-6, 1e-5, 2e-5, 3e-5, 4e-5} |
| **Warm-up Ratio** | 2% | 6% |
| **Learning Rate Decay** | Linear | Linear |
| **Batch Size** | 2048 | 16 or 32 |
| **Sequence Length** | 512 | 512 |
| **Adam $\epsilon$** | 1e-6 | 1e-6 |
| **Adam($\beta_1, \beta_2$)** | (0.9, 0.999) | (0.9, 0.98) |
| **Clip Norm** | 1.0 | 1.0 |
| **Dropout** | 0.1 | 0.1 |
| **Weight Decay** | 0.01 | 0.01 |

and the weight decay are set to 0.1 and 0.01, respectively. The batch size is set to 64. The feature map dimension is set to 64. We use Adam [19] as the optimizer, and set its hyperparameter $\epsilon$ to $1e - 6$ and $(\beta_1, \beta_2)$ to $(0.9, 0.98)$. The peak learning rate is set to $2e - 3$. The model is trained for 150k steps with a 6k-step warm-up stage followed by an inverse square-root learning rate scheduler. All models are trained on 8 NVIDIA Tesla V100 GPUs.

## A.3 Machine Translation

We train the models on the widely used public dataset IWSLT14 German↔English and French↔English. For each language pair, we use a vocabulary of 10K tokens based on a joint source and target byte pair encoding (BPE) [37]. All the experiments use a Transformer encoder-decoder architecture, with a 6-layer encoder and a 6-layer decoder. We set the embedding dimension to 512 and the number of heads to 4. When kernelized attention is used, the feature map dimension is set to 16 in the encoder and 24 in the decoder. For all the models, label smoothed cross entropy is used as the objective function by setting $\epsilon = 0.1$ [40], and we apply dropout with a ratio 0.3. The batch size is set to be 8192 tokens. When we decode translation results from the model during inference, we set beam size as 5 and the length penalty as 1.2. All the models are trained for 50k steps with 4000 warm up steps and a peak learning rate of $5e - 4$, and an inverse square root learning rate scheduler is used after the warm-up stage. All the models are trained with the Adam optimizer [19], in which hyperparameter $(\beta_1, \beta_2)$ is to be $(0.9, 0.98)$ following [45]. BLEU [29] is used as the evaluation measure of the model performance, and we report the BLEU score of a single checkpoint, without using checkpoint averaging. All models are trained on 4 NVIDIA Tesla V100 GPUs.

## A.4 Image Classification

**Baselines.** In the image classification task, we compare our proposed model with several Transformer-based models using softmax attention, including: 1. *ViT* proposed in [10], which first introduces the idea of handling images as sequences of fixed-sized patches and feeding them into Transformer to perform classification; 2. *DeiT* proposed in [42], which largely improves the performance of ViT using strong data augmentation and knowledge distillation[11].

---

[11]For simplicity, in our model and all baseline models, knowledge distillation is not used.

**Training details.** Our training settings mostly follow [42]. We train all the models on the training set of ImageNet-1K [8]. We adopt a default input image resolution of $224 \times 224$ without finetuning the model on a higher resolution. When training NPRF-DeiT from scratch, we employ the AdamW [25] optimizer for 450 epochs using a cosine decay learning rate scheduler with 20 epochs of linear warm-up. When finetuning PRF-DeiT from DeiT with softmax attention, we employ the AdamW optimizer for 100 epochs using a cosine decay learning rate scheduler with 6 epochs of linear warm-up. For the hyperparameters of the optimizer, we set $\epsilon$ to $1e-8$ and $(\beta_1, \beta_2)$ to (0.9, 0.999). Label smoothed cross entropy is used as the objective function by setting $\epsilon = 0.1$ [40]. A batch size of 1024, an initial learning rate of $1e-5$ and a weight decay of 0.05 are used. We employ all the data augmentation and regularization strategies of [42]. Following [24], we perform image classification by applying a global average pooling layer on the output of the last layer, followed by a linear classifier. All models are trained on 8 NVIDIA Tesla V100 GPUs.

## A.5  Image Generation

We additionally conduct experiments on the ImageNet32 image generation task to demonstrate the effectiveness of our proposed method in the long-sequence regime [5]. In this task, the sequence length is 3072.

**Baselines.** In this task, we compare our proposed NPRF-Transformer with several Transformer-based models as well as several other strong baselines, including Image Transformer [30], PRF-Transformer (called Performer in [4]); ScoreFlow [39], VDM [20] and DenseFlow [12].

**Training details.** We train a 6-layer NPRF-Transformer with RPE. For each layer, the hidden size is set to 512, and the number of attention heads is set to 8. When training our model, we use a batch size of 64. We use Adam [19] as the optimizer, and set its hyperparameter $\epsilon$ to $1e-6$ and $(\beta_1, \beta_2)$ to (0.9, 0.98). We set the dropout ratio and the weight decay to 0.1 and 0.01, respectively. The feature map dimension is set to 32 whenever kernelized attention is used. The peak learning rate is set to $5e-4$. We use the BPD (bits per dim) to evaluate the performance of the models.

**Results.** The result is shown in Table 6. It's easy to see that our NPRF-Transformer with RPE outperforms other Transformer-based models by a large margin. Specifically, it outperforms PRF-Transformer by 0.36 points, which shows that our method largely improves the performance kernelized attention on large-scale dataset. Besides, our NPRF-Transformer with RPE also achieves competitive performeance compared with other strong baselines, and is almost on par with DenseFlow, the state-of-the-art model for image generation. These results clearly demonstrate the effectiveness of our proposed method in long-sequence regime.

Table 6: Bits per Dimension (Bits/Dim) on ImageNet32. "*" indicates the best model and "**" indicates the best Transformer-based model.

| Method | BPD |
|---|---|
| ScoreFlow [39] | 3.76 |
| VDM [20] | 3.72 |
| DenseFlow [12] | 3.63* |
| Image Transformer [30] | 3.77 |
| PRF-Transformer | 4.04 |
| NPRF-Transformer with RPE (Ours) | 3.68** |

# B Proofs

## B.1 Proof of Proposition 1

*Proof.* The proof is done by contradiction.

Equation 7 implies

$$\exp(\alpha_{ij}^{rel} - \alpha_{ij}^{vanilla}) = \frac{\sum_{k=1}^{n} \exp(\alpha_{ik}^{rel})}{\sum_{k=1}^{n} \exp(\alpha_{ik}^{vanilla})}, \quad \forall i, j. \tag{15}$$

Thus, $\alpha_{ij}^{rel} - \alpha_{ij}^{vanilla}$ only depends on $i$, and we assume $\alpha_{ij}^{rel} - \alpha_{ij}^{vanilla} = \beta_i$.

Let $X = \left[ x_1^\top, \cdots, x_n^\top \right]^\top \in \mathbb{R}^{n \times d}, B = \{b_{i-j}\}_{i,j=1}^n, \boldsymbol{\beta} = [\beta_1, \cdots, \beta_n]^\top$. Then we obtain

$$\frac{1}{\sqrt{d}} X \left( W_{rel}^Q (W_{rel}^K)^\top - W_{vanilla}^Q (W_{vanilla}^K)^\top \right) X^\top + B = \boldsymbol{\beta} \mathbf{1}^\top \tag{16}$$

$$\Rightarrow B = \frac{1}{\sqrt{d}} X \left( W_{vanilla}^Q (W_{vanilla}^K)^\top - W_{rel}^Q (W_{rel}^K)^\top \right) X^\top + \boldsymbol{\beta} \mathbf{1}^\top. \tag{17}$$

Note that $rank \left( X \left( W_{vanilla}^Q (W_{vanilla}^K)^\top - W_{rel}^Q (W_{rel}^K)^\top \right) X^\top \right) \leq d, rank \left( \boldsymbol{\beta} \mathbf{1}^\top \right) \leq 1$, thus the above equation implies $rank(B) \leq d + 1$, which cannot hold if $B$ is full-rank since $n > d + 1$. $\square$

## B.2 Proof of Theorem 3

*Proof.* By definetion we have

$$A = \left( \frac{\exp(qk_j^\top)}{\sum_{j'=1}^{n} \exp(qk_{j'}^\top)} \right)_{j=1}^{n}; \hat{A} = \left( \frac{\phi(q)\phi(k_j)^\top}{\sum_{j'=1}^{n} \phi(q)\phi(k_{j'})^\top} \right)_{j=1}^{n}. \tag{18}$$

For fixed $q$ and $k$ and any $\varepsilon > 0$, applying Chebyshev's Inequality and lemma 2 we obtain

$$\Pr \left( |\exp(qk^\top) - \phi(q)\phi(k)^\top| \geq \varepsilon \exp(qk^\top) \right) \leq \frac{\exp(\|x + y\|^2) \left( 1 - \exp(-\|x + y\|^2) \right)}{m\varepsilon^2} \tag{19}$$

$$\leq \frac{\exp(\|x + y\|^2)}{m\varepsilon^2} \tag{20}$$

$$\leq \frac{\exp(2\|x\|^2 + 2\|y\|^2)}{m\varepsilon^2} \tag{21}$$

$$\leq \frac{\exp(4R^2)}{m\varepsilon^2}. \tag{22}$$

Applying union bound for $(q, k_1), \cdots, (q, k_n)$, and setting $m = \frac{n \exp(4R^2)}{\varepsilon^2 \delta}$ we obtain

$$\Pr \left( \exists j, |\exp(q \cdot k_j) - \phi(q) \cdot \phi(k_j)| \geq \varepsilon \exp(q \cdot k_j) \right) \leq \frac{n \exp(4R^2)}{m\varepsilon^2} = \delta. \tag{23}$$

Assume $\forall j, |\exp(qk_j^\top) - \phi(q)\phi(k_j)^\top| < \varepsilon \exp(qk_j^\top)$. Then $\forall j$

$$\frac{1 - \varepsilon}{1 + \varepsilon} \cdot \frac{\exp(qk_j^\top)}{\sum_{j'=1}^{n} \exp(qk_{j'}^\top)} < \frac{\phi(q)\phi(k_j)^\top}{\sum_{j'=1}^{n} \phi(q)\phi(k_{j'})^\top} < \frac{1 + \varepsilon}{1 - \varepsilon} \cdot \frac{\exp(qk_j^\top)}{\sum_{j'=1}^{n} \exp(qk_{j'}^\top)} \tag{24}$$

$$\Rightarrow \left| \frac{\phi(q)\phi(k_j)^\top}{\sum_{j'=1}^{n} \phi(q)\phi(k_{j'})^\top} - \frac{\exp(qk_j^\top)}{\sum_{j'=1}^{n} \exp(qk_{j'}^\top)} \right| < \frac{2\varepsilon}{1 - \varepsilon} \cdot \frac{\exp(qk_j^\top)}{\sum_{j'=1}^{n} \exp(qk_{j'}^\top)} \tag{25}$$

$$\Rightarrow \|A - \hat{A}\|_1 < \frac{2\varepsilon}{1 - \varepsilon} < 4\varepsilon. \tag{26}$$

Therefore, $\Pr \left( \|A - \hat{A}\|_1 \geq 4\varepsilon \right) \leq \delta$, which completes the proof. $\square$