# OpenReview forum: "Stable, Fast and Accurate: Kernelized Attention with Relative Positional Encoding"
_NeurIPS.cc/2021/Conference — NeurIPS 2021 Poster_

### Official Review · Reviewer_npFH · 2021-07-14

**Rating:** 7
**Confidence:** 4

**Summary:**

There are two major contributions of this paper. First, it proposes a method to incorporate relative positional embeddings (RPE) into kernelized attention, which is based on Fast Fourier Transformation (FFT) and Toeplitz matrix multiplication and leads to a $O(n \log n)$ time complexity compared to $O(n^2)$ using the conventional self-attention with RPE. Second, it proposes to normalize the queries and keys by their `$\ell_2$ norms, which makes the optimization more stable.

**Limitations And Societal Impact:**

While I believe reducing the time complexity is impactful, the impact of the paper is sort of limited by the experimental results it presents for the following two reasons:
1. The authors only show the forward speed in Fig. 1 (a). In all tables in Sec. 5, the authors only show the time complexity instead of the real inference time on the real datasets. This worries me that this paper may be yet another paper with great time complexity but relatively slow inference time on real use cases.
2. While the paper is proposing an efficient alternative to self-attention, The authors do not show the performance-efficiency trade-off. I wish I could see that the method is at the Pareto frontier. For example, showing with a similar inference speed, this method outperforms linear attention and softmax attention. Linear attentions have $O(n)$ time complexity, so it fairer to compare with a larger model with linear attention and the same inference time as the proposed PRF + RLE method. Similarly, comparing the proposed method to a smaller softmax attention model with the same inference time is also expected.

I am willing to raise my score to 8 if both of these concerns are addressed.

**Main Review:**

First, I summarize the pros and cons of this paper.

Pros:
1. The proposed method is simple and novel.
2. Adding the proposed RPE improves the performance over the linear kernelized attentions.
3. The authors conduct experiments on a variety of tasks: language pre-training, image classification, and machine translation and show that the proposed method is more accurate than the kernelized attention baselines.
4. The authors demonstrate that the proposed normalized kernelized attention is not more more accurate and more stable.


Cons:
1. No inference time provided on real datasets (See Limitations And Societal Impact for details)
2. Not showing the performance-efficiency tradeoff (See Limitations And Societal Impact for details)
3. The proposed method operates at only $O(n \log n)$ time, but the authors do not compare with simple linear baselines which has relative positional information like just using convolutions [1] or a combination of convolutions and kernelized attention. This makes me wonder why we should prefer the proposed $O(n \log n)$ method rather than a $O(n)$ baseline.
4. Figure 3 (b) is very hard to read. It would be better to show y-axis between 30 and 35 or 33 and 35.

Overall, the paper is written in high quality and I recommend accept it; however, there are a couple of critical concerns which hinder me from rating this paper higher.

Reference
[1] Tay, Yi, et al. "Are Pre-trained Convolutions Better than Pre-trained Transformers?." ACL 2021.


**Time Spent Reviewing:**

4

---

> ### Author Response · Authors · 2021-08-07
> **Response to Reviewer npFH**
>
> Thank you very much for supporting our work. We will revise our paper to make the figure clear in the next version. We respond to your comments as below.
>
> **Regarding limitation 1**
>
> a. Our work aims to develop efficient Transformers to handle tasks with long sequences. Similar to many previous long sequence acceleration approaches [1,2,3], there is a length threshold $N^*$ in which the proposed method will enjoy acceleration only when the sequence length is larger than $N^*$. Our $N^*$ is about 1e+3 in practice. For example, in the BERT pre-training task, our model's training time per epoch is similar to that of a standard Transformer. Such experiments validate that our method doesn't hurt the performance compared to the standard Transformer and is better than the baseline methods.
>
> b. To address the concern "this paper may be yet another paper with great time complexity but relatively slow inference time on real use cases.". We follow one reviewer's suggestion to run our model on ImageNet64 in [3], where the standard Transformer cannot handle [3]. We find our model performance is better than [3] at the same training steps in the early stage (Due to the time limitation, we cannot finish the ImageNet64 experiments before the rebuttal deadline. ). We are confident that our final model will still be better as we address the optimization issue in [3] and use the powerful relative positional encoding.
>
>
> **Regarding limitation 2**
>
> To study the performance-efficiency trade-off, we follow your suggestion and conduct an experiment on the IWSLT de-en machine translation task. In particular, we train an 8-layer encoder-decoder PRF model, which has comparable speed with our model. However, it only achieves a 32 test BLEU score, underperforming our model by more than two points. As for comparisons with vanilla Transformer, we have demonstrated in section 4.2 that our model achieves comparable performance with vanilla Transformer of the same size (34.7 v.s. 34.3).
>
> We further kindly point out that the performance-efficiency trade-off can only be evaluated when the models can be successfully trained. On the large-scale language pre-training, even training a base-sized PRF Transformer is almost impossible. It suffers from the optimization instability issue and usually diverges. From this perspective, our model is clearly superior to the baseline models as it can be easily trained from scratch.
>
> **Regarding Cons 3**
>
> Thanks for the reference and suggestions. We found in the reference [4] you mentioned, the model consistently underperforms ours on the shared tasks (GLUE benchmark) as below:
>
> |         | CoLA | RTE  | MRPC | STS-B | MNLI     | QNLI | QQP  | SST-2    |
> | ------- | ---- | ---- | ---- | ----- | -------- | ---- | ---- | -------- |
> | ConvNet | /    | /    | /    | /     | 75.0     | /    | /    | 92.2     |
> | Ours    | 64.7 | 75.1 | 88.5 | 89.4  | **86.4** | 91.3 | 91.7 | **94.8** |
>
> Besides, the model in [4] has much more parameters than ours (230M v.s. our 110M). We believe that this clearly demonstrates that our model is more powerful than convolutional architectures. We can discuss this in the next version of the paper.
>
> We hope our explanations can address your concerns and hope the reviewer can re-evaluate our contributions.
>
> [1] Katharopoulos, Angelos, et al. "Transformers are rnns: Fast autoregressive transformers with linear attention." ICML. 2020.
>
> [2] Peng, Hao, et al. "Random Feature Attention." ICLR. 2021.
>
> [3] Krzysztof, Choromanski,et al. "Rethinking attention with performers." ICLR, 2021.
>
> [4] Tay, Yi, et al. "Are Pre-trained Convolutions Better than Pre-trained Transformers?." ACL 2021.

---

> > ### Comment · Reviewer_npFH · 2021-08-17
> > **Reply to authors' responses**
> >
> > I have read the authors' responses. Limitation 2 and con 3 have been addressed. However, for limitation 1, I still don't see evidence that convinces me. Therefore, I would only raise my rating to 7.5.

---

> > > ### Author Response · Authors · 2021-08-26
> > > **Response to Reviewer npFH**
> > >
> > > Thank you for your feedback. We have been working on additional experiments and will update the empirical results in the final version of our paper.

---

### Official Review · Reviewer_LQ92 · 2021-07-15

**Rating:** 6
**Confidence:** 5

**Summary:**

The paper introduces a novel way to add relative positional encoding to efficient transformers with kernelized attention. The authors make use of the efficient matrix multiplication of Toeplitz matrices using the fast Fourier transform.  The authors perform several experiments on masked language modeling, machine translation and even vision transformers.

**Limitations And Societal Impact:**

The authors do not discuss the societal impact of their work.

The limitations are not adequately discussed as mentioned in the weaknesses section of the main review.

**Main Review:**

Strengths
-------------

- Using the fast multiplication of Toeplitz matrices based on the Fourier transform to implement relative positional encodings is brilliant and very interesting
- The experimental evaluation shows that kernelized transformers with the proposed RPE can actually train and often achieve better results than the same model without RPE
- The theoretical analysis regarding the norm of the queries and keys as well as the ablations and analyses in section 4.4 are great and of independent interest. In particular, I find Figure 2 very informative that concisely showcases both the fact that reducing the norm of the queries and keys helps the random Fourier feature approximations as well as that RPE accounts for a large part of the attention distribution (when used) and helps even more when approximating.
- The authors provide clean code with their submission

Weaknesses
-----------------

### Asymptotic complexity vs real world speed

The main weakness of the paper stems from the misrepresentation of the complexity of the attention computation using the proposed algorithm. The algorithm in its provided form, requires storing in memory a matrix of size $O(ND^2)$ (namely $\phi(K_i) V_i^T$ for all $i$) per sample per head. This is the same issue as showcased in [15] for autoregressive kernelized attention. The result is a theoretically fast algorithm but a slow one in practice.

This can be seen also from Figure 1a. Even at 32k sequence lengths and using only the forward pass, the proposed method needs to down-project the queries and keys to half the dimensions in order to be faster than the **full softmax** attention. In comparison, the simplest kernelized attention by [15] is **10x times** faster in the same experiment. Down-projecting using PRF could make it even faster.

Finally, evaluation of the computational cost is not provided even for a single one of the real world experiments. For instance what is the computational cost of an epoch in GPU wall clock time for the models in tables 1, 2 and 3. Since the proposed model is an approximation of the softmax transformer it should at least be verified that it is faster than that.

### Minor comments

1. The wording and notation in section 3.2 is unnecessarily complicated. The $vec(\cdot)$ notation can be omitted and if needed intermediate matrices can be properly defined together with their dimensions.

**Time Spent Reviewing:**

8

---

> ### Author Response · Authors · 2021-08-07
> **Response to Reviewer LQ92**
>
> Thank you very much for supporting our work. We will revise our paper to address the writing issues you mentioned in the "Minor comments" paragraph.
>
> We noticed that your major concerns are two-fold: the dependency in $D$ of our model and the use of our model in real-world tasks.
>
> **Regarding the first concern**: it is worth noting that our NPRF Transformer w/ RPE can achieve good performance when the feature map dimension $D$ is significantly small, so the dependency in $D$ is not a big issue. To see this clearly, we take the IWSLT de-en machine translation task as an example. For our NPRF Transformer w/ RPE, the model performance is 34.3 even when the feature map dimension $D=16/24$. See Section 4.2 in our paper. Furthermore, the performance only drops a little even when we set $D=8$ (**16x down-projection**). See Figure 3(a) in our paper. On the other hand, for the Linear Transformer [1] mentioned by the reviewer, the final BLEU score is 29.9 even when $D=128$, see [2].
>
> The experimental results above clearly show that our model is memory-efficient as we can use a small $D$ and performs well. We achieve this by using a better kernel approximation [2,3] and relative positional encoding and addressing the optimization problem properly.
>
> **Regarding the second concern**: it is a good catch. We use asymptotic complexity with respect to the length $n$. Similar to many previous long sequence acceleration approaches [1,2,3], there is a length threshold $N^*$ in which the proposed method will enjoy acceleration only when the sequence length is larger than $N^*$. As we can use a small $D$ to get a good performance, our $N^*$ is about 1e+3 in practice. For example, in the BERT pre-training task, our model's training time per epoch is similar to that of a standard Transformer. For extremely short downstream tasks, such as CoLA, where the average sequence length is only about 20, our fine-tuning time per epoch is longer than that of a standard Transformer (0.6x).
>
> Our proposed model makes improvement when the sequence is very long. We follow one reviewer's suggestion to study an extreme case by running our model on ImageNet64 following [3], where the standard Transformer cannot handle [3]. We find our model performance is better than [3] at the same training steps in the early stage (Due to the time limitation, we cannot finish the ImageNet64 experiments before the rebuttal deadline. ). We are confident that our final model will still be better as we address the optimization issue in [3] and use the powerful relative positional encoding.
>
> We hope our explanations can address your concerns and hope the reviewer can re-evaluate our contributions.
>
> [1] Katharopoulos, Angelos, et al. "Transformers are rnns: Fast autoregressive transformers with linear attention." ICML. 2020.
>
> [2] Peng, Hao, et al. "Random Feature Attention." ICLR. 2021.
>
> [3] Krzysztof, Choromanski,et al. "Rethinking attention with performers." ICLR, 2021.

---

### Official Review · Reviewer_AuHp · 2021-07-16

**Rating:** 7
**Confidence:** 4

**Summary:**

The authors propose a method to compute kernelized attention with relative position encoding (RPE) efficiently in $O(n\log n)$. They show that the RPE family of attention is richer than attention without RPE, and the usual $O(n)$ implementation of kernelized attention is no longer possible.  The derivation gives rise to a convolution, which the authors propose to compute in the Fourier domain, providing $O(n\log n)$ complexity, in contrast to $O(n^2)$  for a naive implementation.

The paper provides a comprehensive derivation of the algorithm, mathematically and in pseudo code. Interestingly, it is found that with the help of RPE training stability of kernelized attention can be improved, thus allowing for the first time to train certain models end-to-end.

Experiments with random toy data where performed to assess the computational load of the algorithm, and experiments on datasets coming from language pre-training, machine translation, and image classification are used to show the effectiveness and the improved training stability of the algorithm on real world problems






**Limitations And Societal Impact:**

The authors don't explicitly address limitations of their model, other than additional speedup for sequence generation is not provided.
Another potential limitation, that should be mentioned, could come from the fact that NPRF even with RPE cannot generate sparse attention to the same extend as PRF or plain dot-then-exponentiate attention  (see above)

Although not explicitly addressed in the paper,  I don't see any negative societal impact of the work.

**Main Review:**

## Originality
The paper brings three well known techniques together: kernelized attention, RPE, and the Convolution Theorem, to enable kernelized RPE attention. It's a nice idea that leads to an elegant algorithm. To my knowledge it is the first time that this has been proposed. All prior work has been cited appropriately.

## Quality
The overall quality of the paper is very good. The derivations are sound and the experiments support the claims made by the authos.


There are two things missing that I think should be discussed:

* How does the model behave with causal attention. I think it can be implemented due to the RPE by setting $c_i=0$ for $i\ge 0$ in (11) and (12). Has this been done in the experiments and found to work? If so, then I think footnote 1 is not a limitation unique to this algorithm. Sequence generation at inference time always gives another factor $n$, and so does it here. The convolution does not improve on that, bat at least it does not make things worth (at least compared to non-kernelized attention)

* For the *normalized kernelized attention with RPE* ( l 203 - 209) it should be noted that this is not the same as *unnormalized kernelized attention*. You can produce sharp distributions but only at specific distances for which for which $b_{i-j}$ happens to be large. So NPRF w/ RPE has lower expressivity than PRF, still showing better performance (Tab. 3). It would add to the strength of the paper if the authors could elaborate on that.

## Clarity
The paper is very well written. The mathematical derivation is comprehensive an easy to follow. All important information is in the main paper

## Significance
RPE have been shown to have a big impact on the performance of attention based models on various problems. On the other hand, kernelized attention can save a big deal of computational cost and make transformer models scalable to real world large scale problems. To link both approaches in an efficient way clearly advances the field and has a big impact for practitioners working with transformer models.

**I vote for accepting the paper.**

Minor issues
* l 79: indicate the axis of the softmax operation
* Sec. 2.1 ff:  all $x_i$ are row vectors which is unusual and can lead to confusion, in particular when writing $x=(x_1,\ldots,x_n)$. You should either revise the notation or state explicitly that $x_i\in\mathbb R^d$ is a row vector
* l 88 - 94: wherever you have running variables $j'$ in summations, make sure to also use $j'$ in the summation terms.
* Fig. 1a would gain from one more sequence length step ($n=48k$ or $n=64k$) showing the (almost) linear relation better




**Time Spent Reviewing:**

4

---

> ### Author Response · Authors · 2021-08-07
> **Response to Reviewer AuHp**
>
> Thank you very much for supporting our work and careful review! We will revise our paper to address the writing issues you mentioned in the "Minor issues" paragraph.
>
> **Regarding the causal attention:** It is correct that our method can be used in the casual setting. We actually have conducted experiments on language modeling tasks (where causal attention is used) and achieved better performance than the baselines. Due to space limitations, we put it in Appendix A.1 in the attached supplementary material.
>
> **Regarding the expressiveness:** This is a good catch. We agree that the expressive power of "normalized" PRF Transformer w/ RPE can be weaker than Transformer as it can only produce context-agnostic sharp distribution (from RPE) but may not produce context-aware sharp patterns. However, this does not conflict with the observation that the "normalized" PRF Transformer w/ RPE achieves better performance than PRF. The performance of a learned model depends on not only the expressive power but also the optimization difficulty. Although PRF Transformer seems to have more flexibility, it has the training instability issue, making the model difficult to learn in practice.
>
> Besides, empirically, the loss of expressiveness ("normalized" PRF Transformer w/ RPE v.s. Transformer) hurts the performance little. For example, on the image classification task, it only underperforms vanilla Transformer by no more than 0.3 points in terms of top-1 and top-5 accuracy. That being said, the model is still expressive enough to capture the essential information in learning real-world tasks.

---

> > ### Comment · Reviewer_AuHp · 2021-08-24
> > **Thanks for tour response.**
> >
> > I'm happy to hear that you intend to address all the minor issues I came up with.
> >
> > As for causal attention, I couldn't find any background on how to use it in a causal setting. You're presenting a language modelling task, so apparently you have been using it like that, but the how is missing. Is it sufficient to clip $D_1$ and $D_2$ to triangular matrices? It would be very helpful to drop a note on that in the main paper.
> >
> > As for the expressiveness, this reads to me as if the RPE introduces an inductive bias that helps with certain problems (and certainly hurts others). I think that's an interesting line of though. You should perhaps pick it up in the discussion.

---

> > > ### Author Response · Authors · 2021-08-26
> > > **Response to Reviewer AuHp**
> > >
> > > Thank you for your feedback. For the causal language modeling, we only need to set $b_{i-j}=-\infty$ (i.e., $c_{i-j}=0$) when $i>j$. Note that the positional-correlation matrix $C$ is still a Toeplitz matrix, so our method also works in this setting.
> > >
> > > Thanks for your suggestions. We will discuss the causal language modeling implementation and the model's expressiveness (with RPE) in the main body of the paper.

---

> > > > ### Comment · Reviewer_AuHp · 2021-09-10
> > > > **Thanks for your responses.**
> > > >
> > > > I think that all my issues have been addressed.

---

### Official Review · Reviewer_mmJ4 · 2021-07-16

**Rating:** 8
**Confidence:** 3

**Summary:**

This paper contributes to the literature on efficient [transformer][]
architectures. Specifically, papers that focus on reducing the quadratic
computational complexity of the self-attention mechanism between tokens.

The problem this paper focuses on is the incompatibility of [relative
positional encodings (RPE)][rpe] and sub-quadratic efficient attention
mechanisms. It adapts the kernelized attention used in the [Performer][]
architecture to allow the use of RPE. In the process, the authors find that
this adaptation allows use of the FFT algorithm to perform the approximate
attention mechanism, resulting in an $O(N \log N)$ algorithm, where $N$ is
the number of tokens.

Benchmarks are provided on the GLUE dataset according to various metrics
and on the ImageNet dataset using Image Transformer variants. Performance
to competing methods chosen is better on most of the metrics presented.

[transformer]: https://papers.nips.cc/paper/2017/file/3f5ee243547dee91fbd053c1c4a845aa-Paper.pdf
[rpe]: https://arxiv.org/abs/1803.02155
[performer]: https://arxiv.org/abs/2009.14794

**Limitations And Societal Impact:**

Efficient self-attention methods with sub-quadratic computational
complexity are typically applied to problems that have a large number of
tokens. An example of this can be seen in the experiment in Section 4.5 of
the [Performer][] paper, on the ImageNet64 task that cannot be addressed by
traditional transformers. This paper does not include experiments with
numbers of tokens that would be infeasible for a normal transformer.

The authors discuss how normalizing the kernelized attention reduces the
capacity of the model to express sharp attention distributions in the final
paragraph of Section 3.3. As the goal of this work is not to lose the extra
capacity offered by [RPE][] when using kernelized attention it is a
limitation that the authors do not present an experiment investigating how
the model performs without normalization and with RPE (ie PRF w/ RPE in the
notation of the paper's experiments). It may be necessary to address the
stability using the same tricks employed by [Performer][] to run this
experiment.

The FFT algorithm proposed to perform the Toeplitz matrix multiplication
receives little discussion, save that it is possible. I think this is
appropriate but there are some concerns that could be addressed:

1. FFT algorithms are not well supported in all deep learning frameworks
and FFT Toeplitz matrix multiplications are not supported directly
2. FFT algorithms, unless carefully implemented, require the inputs to be a
power of 2 number of dimensions
3. Implementing this using FFT operations in an autograd system will have a
relatively large number of intermediate variables that will be stored for
the backward pass and could increase memory usage. This could be addressed
with custom operations that define the appropriate backward pass directly.

[performer]: https://arxiv.org/abs/2009.14794
[rpe]: https://arxiv.org/abs/1803.02155

**Main Review:**

## Originality

> Are the tasks or methods new? Is the work a novel
> combination of well-known techniques? (This can be valuable!) Is it clear
> how this work differs from previous contributions? Is related work
> adequately cited?

This work is a novel combination of two existing methods: [relative
positional encodings][rpe] and [kernelized efficient
self-attention][performer]. In the process they make an independent
contribution, discovering that combining the two methods enables use of the
FFT algorithm to compute the efficient self-attention mechanism. This fills
its own niche among the many existing efficient self-attention mechanisms.

Related work cited is a thorough review of efficient self-attention
methods, with each correctly sorted into their respective types. The review
makes it clear where this method fits into this set.

In addition the authors make various observations that will be useful to
machine learning practitioners:

1. Ablation comparisons of [Performer][] and [Random Feature
Attention][rfa]
2. Analysis of the function class of transformers using [RPE][]
3. Analysis of the approximation error of kernelized attention
    1. Fourth paragraph of Section 3.3 makes a useful observation about the
training dynamics of transformers using kernelized attention
    2. This is further put into context in the literature by the footnotes
in the following paragraph on "Normalized Kernel Attention with RPE"
4. Address the limitations of existing kernelized attention mechanisms,
such as [Performer][], that are unstable when trained from scratch by using
a normalization step

[rpe]: https://arxiv.org/abs/1803.02155
[performer]: https://arxiv.org/abs/2009.14794
[rfa]: https://openreview.net/forum?id=QtTKTdVrFBB

## Quality

> Is the submission technically sound? Are claims well supported
> (e.g., by theoretical analysis or experimental results)? Are the methods
> used appropriate? Is this a complete piece of work or work in progress? Are
> the authors careful and honest about evaluating both the strengths and
> weaknesses of their work?

All claims in the paper are supported by theoretical inquiry and
experiment. The experimental results presented demonstrate the claims
made in the text. On the benchmarks chosen (GLUE, machine translation,
ImageNet) the network performs well in comparison to the methods chosen for
comparison.

The technical analysis is in depth and considers a limitation of the work.
In a footnote of Section 3.2, the authors mention that the method cannot be
used to improve the efficiency of the inference step. In practice the
memory explosion of the kernel matrix during training is a larger problem
when training transformers than the inference step memory usage. But, it
may affect this method's ability to scale to large sequence lengths.

Lemma 2 in particular is an important observation about the variance of
kernelized attention and is backed up in the paper by theoretical analysis
and precise experimental exploration illustrated in Figure 1. Both of these
together are very effective in convincing the reader that this Lemma is
correct and worthwhile.

[performer]: https://arxiv.org/abs/2009.14794

## Clarity

> Is the submission clearly written? Is it well organized? (If not, please
> make constructive suggestions for improving its clarity.) Does it
> adequately inform the reader? (Note that a superbly written paper provides
> enough information for an expert reader to reproduce its results.)

The submission is clearly written and well organised. Reading this paper in
sequence each topic is introduced in a way that is easy to understand and
each idea follows from the previous. I am confident that any researcher
familiar with transformers will be able to understand this paper.

Section 2 deserves praise for providing a concise and useful description of
kernelized attention. It helps to make the contribution of the paper clear
to the reader by drawing together various prior work in kernelized
attention under a common framework. I would use it as a reference if I were
writing about kernelized attention in future.

Some minor suggestions I would make:

1. The related work is discussed in the Introduction, which may not draw
the eye for those searching for it.  However, putting it in it's own
section would ruin the flow and probably push the paper over the page
limit. I would suggest putting a `\paragraph` bold title indicating that
paragraph 2 of the Introduction is the Related Work section but if the
author's find it worse then I would not argue.
2. Section 3 mentions the authors will "develop new techniques" but does
not indicate what those techniques are. It would be better to be precise
about what the new techniques are.
3. Sentence 3 of Section 3.3 has a grammar mistake that impedes
readability, "those kernelized attention" should probably be "kernelized
attention"

## Significance

> Are the results important? Are others (researchers or practitioners) likely
> to use the ideas or build on them? Does the submission address a difficult
> task in a better way than previous work?  Does it advance the state of the
> art in a demonstrable way? Does it provide unique data, unique conclusions
> about existing data, or a unique theoretical or experimental approach?

The specific niche this work fits into in the literature on efficient
self-attention mechanisms is useful. This seems like a natural question
when considering [RPE][] and [kernelized attention][performer].

In addition, the demonstration of how the FFT algorithm may be used to
perform a self-attention mechanism appears valuable. It seems like there
may be ways to build upon it, such as incorporating it with other relative
positional encodings, such as [rotary encodings][rotary] and the result may
continue to improve performance. There are likely other ways this work
could be adopted and improved into Transformer design.

[rpe]: https://arxiv.org/abs/1803.02155
[performer]: https://arxiv.org/abs/2009.14794
[rfa]: https://openreview.net/forum?id=QtTKTdVrFBB
[rotary]: https://arxiv.org/abs/2104.09864

**Time Spent Reviewing:**

4

---

> ### Author Response · Authors · 2021-08-07
> **Response to Reviewer mmJ4**
>
> Thank you very much for supporting our work! We will correct the typos and follow your writing suggestions in the next version.
>
> Thanks for the suggestion on the extremely long-sequence experiment. Due to the time limitation, we cannot finish the ImageNet64 experiments before the rebuttal deadline. We will include it in the next version of the paper once completed. At the current stage of the ImageNet64 training, our model is better than Performer at the same training step. We are confident that our final model will still be better as we address the optimization issue and use the powerful relative positional encoding.
>
> Thanks for the suggestion on trying "unnormalized" PRF Transformer w/ RPE. We have conducted these experiments before the paper submission. On the IWSLT translation task, we found that "unnormalized" PRF Transformer w/ RPE achieved comparable performance with vanilla Transformer and our "normalized" PRF Transformer w/ RPE. However, on the large-scale BERT pretraining task, we observed the training of "unnormalized" PRF Transformer w/ RPE sometimes diverged, even with the Performer training tricks (e.g., orthogonal kernel). We conjecture it's due to the large variance of PRF's approximation (Section 3.3). We can include the experimental results and discussions in the paper if needed.
>
> We will add discussions on FFT algorithms in the next version of the paper. It is worth noting that FFT algorithms have become more and more popular in deep learning [1,2,3], and the modern deep learning frameworks are putting effort to support FFT algorithms. For example, PyTorch 1.7 added a new `torch.fft` module that implements FFT-related functions. Based on these APIs, we implemented the FFT Toeplitz matrix multiplication and submitted corresponding codes in the supplementary materials. We believe that our methods can be further accelerated by better implementation in the future.
>
> [1] Tri, Dao, et al. "Learning fast algorithms for linear transforms using butterfly factorizations." ICML. 2019.
>
> [2] Tri, Dao, et al. "Kaleidoscope: An Efficient, Learnable Representation For All Structured Linear Maps." ICLR. 2020.
>
> [3] Lee-Thorp, James, et al. "FNet: Mixing Tokens with Fourier Transforms." arXiv preprint arXiv:2105.03824 (2021).

---

> > ### Comment · Reviewer_mmJ4 · 2021-08-24
> > **ImageNet64 and FFT Implementations**
> >
> > I'm looking forward to seeing the results of the ImageNet64 experiment and I think it should be of interest to the community. The intermediate results against Performer you mention are certainly exciting. It is a shame such large computational resources are required to compare against contemporary works in this field.
> >
> > The FFT algorithm implementations on their own will also be a useful contribution to the field. In my own experience and talking to other researchers people are often in the position of finding good reference implementations of algorithms involving FFTs, even using modules like `torch.fft` it can be tricky to implement in a way that is more efficient due to intermediate activation storage with large activation tensors. I haven't looked at your implementation yet, did you consider using the inverse FFT on the backward pass to recompute activations in order to save memory?

---

> > > ### Author Response · Authors · 2021-08-26
> > > **Response to Reviewer mmJ4**
> > >
> > > Thank you for your feedback. We will update the experiment results in the final version of our paper. In our submitted codes, only the FFT-based Toeplitz matrix multiplication is implemented, while the internal implementation of the ``torch.fft`` module is not modified. Your suggestion on the memory-efficient backward pass by using the inverse FFT is valuable and promising. We will try it and evaluate the time and memory trade-off on further experiments.

---

### Decision · Program_Chairs · 2021-09-27

**Decision:**

Accept (Poster)

**Comment:**

This is an interesting paper extending linear attention mechanisms proposed in Performers to work with an *arbitrary* relative position encoding (RPE) mask. Previous results showed that it is possible for special instantiations of RPE. This paper shows that as long as a mask has Toeplitz structure (this can be actually further extended to a low displacement rank structure), efficient computation of the attention module can be achieved by combining random feature map tricks with Fast Fourier Transformer mechanism (every RPE is by definition a Toeplitz matrix). The authors furthermore claim, that by incorporating RPE into linear attention modules, they can substantially improve their overall accuracy. According to them, this is possible since by adding RPEs, one can constrain queries and keys to be L2-normalized (resulting in the substantial reduction of the variance of the random feature map estimator that grows exponentially in the lengths of keys/queries). Normally such a normalization would affect the expressiveness of the model, but according to the authors, the RPE (which in principle is not bounded) provides extra expressiveness that would be lost otherwise.
This claim is controversial and additional detailed experiments showing that RPE can indeed carry out the otherwise-lost expressiveness would further strengthen the paper (in particular an extensive ablation studies over L2-normalized variants with and without RPE would provide extra clarity).  The paper would also benefit from providing more theoretical understanding of this phenomenon.
The other important point that should be discussed in more detail is an efficient implementation on TPUs of Fast Fourier Transform. Unless such an implementation is provided, practical usage of this algorithm will be limited.
Nevertheless, it is an important result in the field and an elegant approach to incorporating structured masks into kernelizable attention modules. This algorithmic observation is the main contribution of the paper.